



# Evapotranspiration evaluation by 3 different protocols on a large green roof in the greater Paris area

Pierre-Antoine Versini[1,*], Leydy Alejandra Castellanos-Diaz[1,a], David Ramier[2], Ioulia Tchiguirinskaia[1]

[1] HM&Co, Ecole des Ponts, Champs-sur-Marne, 77455, France
[a] formerly at : HM&Co, École des Ponts, Champs-sur-Marne, 77455, France
[2] TEAM, Cerema, Trappes, 78197, France

*Correspondence to*: Pierre-Antoine Versini (pierre-antopine.versini@enpc.fr)

**Abstract**. Nature-Based Solutions have appeared as relevant solutions to mitigate urban heat islands. To improve our knowledge on the assessment of this ecosystem service and the related physical processes (evapotranspiration), monitoring campaigns are required. It was the objective of several experiments carried out on the Blue Green Wave, a large green roof located at Champs-sur-Marne (France). Three different protocols were implemented and tested to assess the evapotranspiration flux at different scales. The first one was based on the surface energy balance (large scale). The second one was carried out by an evapotranspiration chamber (small scale). The third one was based on the water balance evaluated during dry periods (punctual scale). In addition to these evapotranspiration estimates, several hydro-meteorological variables (especially temperature) were measured. Related data and Python programs providing preliminary elements of analysis and graphical representation have been made available. They illustrate the space-time variability of the studied processes regarding their observation scale. The dataset (Versini et al., 2023a) is available here: https://doi.org/10.5281/zenodo.8064053

## 1- Introduction

Nature-Based Solutions (NBS) are solutions that can restore natural processes, while providing a variety of valuable environmental, economic and social benefits (European Commission. Directorate General for Research and Innovation., 2015). Among NBS, green roofs are often proposed for sustainable urban adaptation. They have been widely recognized to reduce urban stormwater runoff (Ayata et al., 2011; Stovin et al., 2012; Versini et al., 2020) and Urban Heat Island (UHI) phenomenon (Cascone et al., 2018; Coutts et al., 2013; Sharma et al., 2016b; Wadzuk et al., 2013b), while preserving biodiversity (Duffaut et al., 2022).

The UHI phenomenon is characterized by a difference of air temperature between urban and rural areas, mainly at night (Oke, 1982). According to Sharma et al. (2016) several factors contribute to the UHI. Among them, the conversion of vegetation by impervious surfaces is pointed, which reduces cooling rates from vegetation evapotranspiration (ET) process. Several studies have found that buildings' roofs represent around 25% of land surface in urban areas (Cascone et al., 2018; Yang et al., 2018). Therefore, the implementation of vegetated structures like green roofs into new constructions or retrofitted ones is highly promoted as a solution to mitigate UHI.



Consequently, understanding the ET process is key to identifying the conditions in which green roofs provide the highest cooling benefit. ET is a combined process by which water is transferred to the atmosphere by evaporation from surfaces and by transpiration from the plants. The evaporation is the physical process of change from liquid water to vapour, while the transpiration is the transfer of liquid water from the plant root system to the leaves, where water is evaporated. The transformation of liquid water to vapour requires sufficient energy to overcome liquid-phase intermolecular forces. In consequence, the heat (i.e., as a form of energy) is removed from the atmosphere because of ET.

In green roofs, mass and heat transfer of water vapour involved in the ET process makes air becoming moist, and reduces surface temperature of the roofs and the nearest environment. Different conditions can affect the ET efficiency of green roofs. These can include external conditions (e.g., local climate, surface albedo, soil moisture, etc.) or internal factors (e.g., substrate and vegetation properties). External conditions as local climate determines the amount of energy available for ET, whereas internal factors govern the water retention and water loss rate. Since a large number of physical, meteorological and vegetation parameters are involved in the ET process of green roofs, several methods for ET measurement have been developed at different scales.

One of the most widely applied methods to measure ET over green roofs is the weighing lysimeter, which is considered as the only direct quantitative technique of ET estimation (Rana and Katerji, 2000). This method monitors weight changes in the green roof structure because water losses from ET. For technical and financial reasons, this method is limited to small prototypes of few m$^2$ (Cirkel et al., 2018; DiGiovanni et al., 2013; Wadzuk et al., 2013a). Laboratory set-up experimentations carried out by Ayata et al. (2011) to estimate ET from weighing lysimeters and water balance methods demonstrated the main advantage of lab experiments is the full control of environmental conditions (e.g., temperature, humidity, solar radiation). Some cheaper devices than lysimeters to monitor green roofs' weight change have also been used. For example, in New Zealand, mass changes due to water loss by ET of eight green roof trays were monitored with a weight sensor (single beam load cells) (Voyde et al., 2010). Besides, experimentations of mass balance and soil moisture variations in green roof plots (laboratory scale) inside a climate chamber were implemented in other studies (Poë et al., 2015; Tabares and Srebric, 2012).

Plant physiology approaches, such as chamber systems have been used on green roofs too. In New York, an experimentation used a portable dynamic closed chamber to survey the water vapor concentration on extensive green roofs (Marasco et al., 2015). Likewise, measurements from a chamber during warm and sunny days in Melbourne (Coutts et al., 2013a) demonstrated the link between the soil moisture and ET.

Moreover, micrometeorological approaches of ET used at large scales in agricultural experimentation, such as the Eddy Covariance (ECO) have been deployed in urban environments too. The ECO technique derives convective fluxes from the covariance of fluctuations in the vertical wind velocity and the atmospheric scalars by turbulent eddies. In consequence, its utilisation has been recognized as a useful method to investigate green roofs and surrounded atmosphere energy exchanges. Experiments conducted with ECO during a year on a green roof located in the California Academy of Sciences, to characterise annual variability of the Surface energy Balance (SEB) parameters (Thorp, 2014), showed the sensible heat flux was the SEB's dominant component during the daytime, meanwhile the latent heat (proportional to ET) was only higher in the first daytime hours. The ECO application by





Heusinger & Weber (2017) over an unirrigated extensive green roof of 8600 m² in Berlin, confirmed the dominant fraction of sensible heat in the warmer periods.


Apart from ECO, additional micrometeorological approaches frequently used in agriculture to deduce latent heat flux by means of the SEB imply the estimation of the sensible heat by scintillometry. This technique is deployed to measure the fluctuations of the air reflective index, induced by temperature changes over a horizontal path. However, because of its implementation

limitation over small horizontal distances, this method has not yet been used in green roofs. Optical scintillometry methods have been recognized by various authors as an accurate and suitable tool to estimate turbulent fluxes, such as the latent heat flux, over homogeneous and heterogeneous surfaces (Guyot et al., 2009; Meijninger et al., 2002; Moene, 2003; Moorhead et al., 2017; Valayamkunnath et al., 2018; Yee et al., 2015).


Due to the wide variability of methodologies and conditions of measuring the ET, the misunderstanding about the ET and the thermo-physical performance of green roofs persists. Moreover, despite this large number of previous studies dedicated to microclimate mitigation by NBS, no data are currently available. In this context, the aim of this paper is to present and

to make available the data collected from 3 ET measurement methods, different by their spatial and temporal scales. These 3 experimental methods were conducted on an extensive wavy-shape green roof, called Blue Green Wave (BGW). They rely on: i) the surface energy balance (SEB) to deduce latent heat fluxes (from measurements of sensible heat flux by the scintillometry technique; ii) the variation of absolute humidity within an evapotranspiration

chamber, and iii) the analysis of water balance during dry periods, to deduce ET by means of water content loss in the ground. Additional hydro-meteorological data characterizing the conditions of the measurement campaigns are also made available: incident and reflected solar and atmospheric radiations, substrate, surface and air temperature, water content, structure parameter of the refractive index of air…

## 2- Materials and methods

### 2.1 The Blue Green Wave

The experimental campaigns were carried out on the Blue Green Wave (BGW), a non-irrigated extensive wavy green roof of 1 ha (see Figure 1) located in front of Ecole des Ponts (ENPC) at Cité Descartes (Champs-usr-Marne, 20 km East of Paris), the research and education core

dedicated to sustainable cities of eastern Paris. The vegetation of the BGW includes mainly green grass and a mix of perennial planting, grasses and bulbous. The depth of the substrate layer varies from 20 cm and 28 cm, depending on vegetation type. The substrate is a composition of volcanic soil completed by organic matter (around 13% in mass). See Stanic et al. (2019) and Versini et al. (2020) for more details on the different layers composing the BGW.




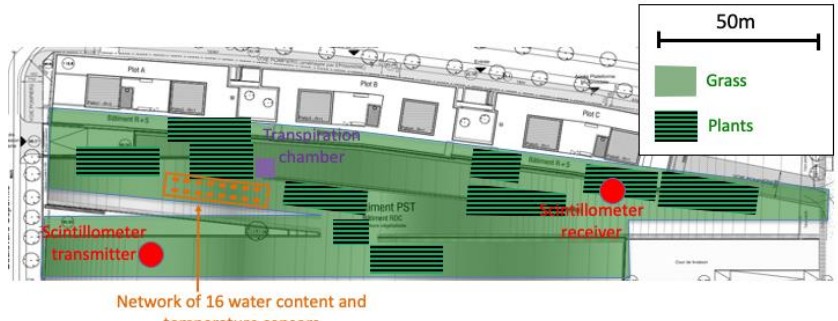

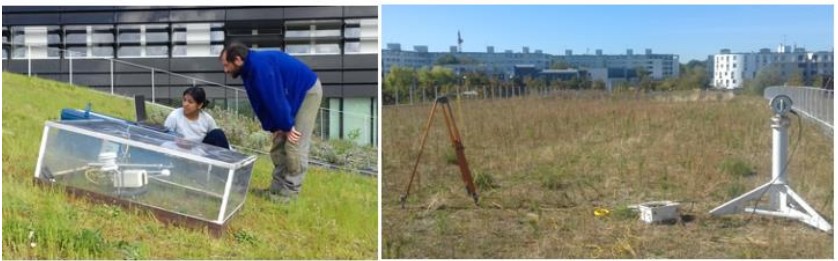

**Figure 1. The Blue Green Wave experimental site: implementation of the different sensors (top), the evapotranspiration chamber (bottom-left), and the surface energy balance instruments (bottom-right)**

Since 2013 and the EIT Climate-KIC Blue Green Dream Project (Maksimovic et al., 2013), the BGW has become an experimental site to study and understand the hydrological and thermal behaviour of extensive green roofs. A permanent monitoring of water balance components (e.g., rainfall, soil water content and temperature, and run-off) has started in 2018 and was presented in a previous data paper (Versini et al., 2020). The objective was to better understand how such NBS can act as a stormwater management tool. To complement this approach and assess their
performances in urban cooling, energy balance and evapotranspiration campaigns were conducted during summer months from 2018 to 2020. These experiments were conducted at 3 different spatial scales which are presented in the following.

## 2.2 Experiment protocols and study scale

### 2.2.1 Large scale assessment by Surface Energy Balance

On green roof surfaces, most radiative energy comes from solar short-wave radiation and, to a lesser degree, from long-wave radiation provided by the ground, the leaves, and the sky. This energy can be partially absorbed or reflected by the vegetation and the ground of the roof. Thus, the net radiative flux ($Rn$) is the difference between the incident and reflected radiative fluxes. $Rn$ represents the main input of the Surface Energy Balance (SEB), which is exchanged with
the green roof, the surrounding atmosphere, and the building structure (see Figure 2) through: the sensible heat flux ($Qh$) produced by convection, the latent heat flux ($Qe$) due to evapotranspiration from vegetation and soil, and the heat conduction into the soil substrate ($Qg$) (De Munck et al., 2013; Marasco et al., 2015). In this way, the energy balance on a green roof can be defined as follows:


$$Rn = Qh + Qe + Qg \qquad (1)$$



All terms of the energy balance are expressed in terms of energy transfer per unit area (W/m$^2$).
The terms of right-hand side of Eq. (1) can be either positive or negative, when they represent
losses or gains of heat for the surface, respectively. By opposite, the sign of $Rn$ is positive when
there is a gain and negative when a loss of heat (Oke, 1987). Additional heat fluxes transferred
by advection, anthropogenic sources, photosynthesis process and heat stored in the substrate
matrix are neglected.

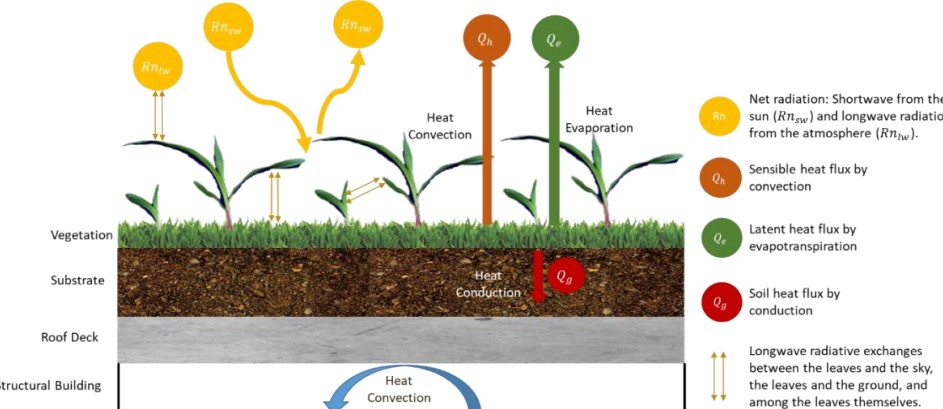

**Figure 2. Surface Energy Balance on a green roof.**

As $Rn$, $Qh$ and $Qg$ are respectively measured by some radiometers, scintillometer and
thermocouples (see next section), $Qe$ can be deduced like the residual component of SEB:

$$Qe = Rn - Qh - Qg \tag{2}$$

### 2.2.2 Small scale assessment by evapotranspiration chamber

A portable evapotranspiration chamber built by the Cerema (*Centre D'études Et D'expertise*
*Sur Les Risques, L'environnement, La Mobilité Et L'aménagement*) was punctually installed
over the BGW vegetation to measure $Qe$. This device (see Figure 1) consists of a one square-
meter enclosed chamber of Plexiglas of 0.3 m$^3$ (1 m$^2$ x 0.3 m) total volume. Gas exchanges,
specifically water vapour (absolute humidity) $H_2O$ and carbon dioxide $CO_2$, were monitored
inside the chamber through a LI-COR LI-7500 gas analyser. The chamber was also equipped
with two small rotating fans (to homogenise the air sample inside the chamber), two T107
temperature sensors and a NR-Lite radiometer (Kipp & Zonen), in order to control microclimate
variation parameters inside the ventilated chamber.

The latent heat flux rate of the volume enclosed by the chamber is deduced from the rise of the
absolute humidity concentration (Coutts et al., 2013). Measurement of absolute humidity are
made each second during two minutes. Then a linear regression is carried out on the first minute
of measurement (assumed to be short enough to not generate microclimate changes inside the
chamber and attain the saturation vapour pressure, see Figure 3). Latent heat flux rate is then
obtained by the slope of the linear regression (Ramier et al., 2015; Ouédraogo et al., 2023), as
follow:



$$Qe = 10^{-3} L_v \frac{V_{ch}}{A_{ch}} \frac{dq}{dt} \tag{3}$$

where, $Qe$ is the latent heat flux (W/m$^2$), $L_v$ is the latent heat of vaporisation of water, $V_{ch}$ and $A_{ch}$ correspond to the volume and the area of the chamber (0.3 m$^3$ and 1 m$^2$, respectively), $\frac{dq}{dt}$

represents the absolute humidity variation during the first minute of measurement (g/m$^3$s), and $10^{-3}$ is a conversion factor.

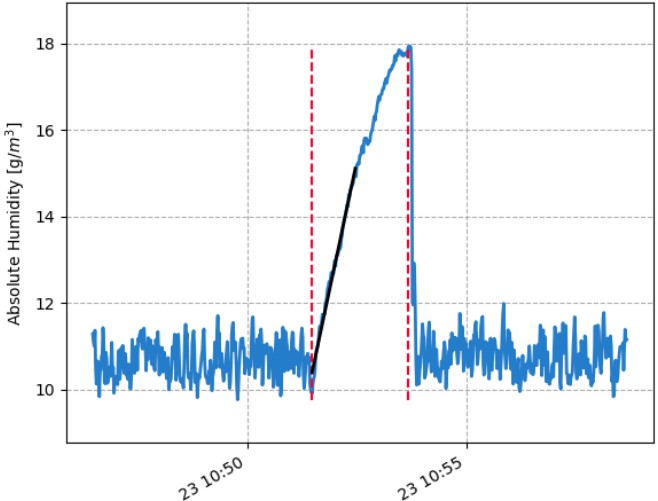

**Figure 3. Time series of absolute humidity variation (blue) for two minutes (star and end time in red dotted lines) and linear regression (black).**

### 2.2.3 Punctual scale assessment by water balance

In a green roof, water input fluxes as precipitation or irrigation can be released to the discharge/sewage network by runoff or to the atmosphere by ET (see Figure 4). Therefore, the

water balance in a green roof can be expressed as follows:

$$P + I = Qr + ET + \Delta S \tag{4}$$

where $P$ represents the precipitation, $I$ the irrigation, $Qr$ is the runoff and $\Delta S$ corresponds to the variation of the stored water in the soil. All the terms of Eq. (4) are expressed in mm/h.

In extensive green roofs, where no irrigation system is usually implemented (i.e., as the BGW case), and during long dry periods without precipitation, there is no water infiltration in the substrate nor runoff. Therefore, for these periods, the ET flux can be estimated from the water balance equation as the soil water content variations:

$$ET = -\Delta S \tag{5}$$

$ET$ can be converted in $Qe$ by:

$$Qe = ETL_v\rho_w \tag{6}$$

where, $L_v$ (2.45×10$^6$ J/kg at 20°C) is the latent heat of vaporisation of water and $\rho_w$ (1000 kg/m$^3$ at 20°C) the water density.

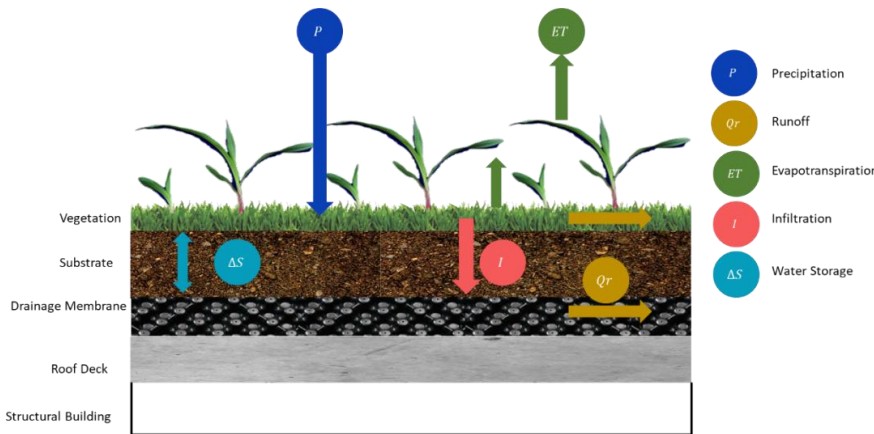


**Figure 4. Water balance for a green roof without irrigation**

## 2.3 Devices

The different sensors used for the 3 measurement protocols are presented in details in this section.

### 2.3.1 Radiometer


To analyse the main components that supply energy to the BGW, a CNR4 radiometer from Kipp&Zonen® (Kipp&Zonen, 2014) was installed close to the scintillometer receiver unit and 1.5 m over the ground. The objective was the measurement of $Rn$, the ratio between the incoming and outgoing short and long-wave radiation (See Eq. (1) and (2)). CNR4 radiometer

includes two pyranometers to measure incident $Sw_{in}$ and reflected $Sw_{out}$ solar or short-wave radiation, and two pyrgeometers to estimate long-wave radiation from the sky $Lw_{in}$ and the ground $Lw_{out}$. Because of the gain of energy for the surface from the incident radiation, $Sw_{in}$ and $Lw_{in}$ are positive, while the energy reflected, $Sw_{out}$ and $Lw_{out}$ is negative. All radiation components are used to calculate the net total radiation on the BGW like this:


$$Rn = (Sw_{in} - Sw_{out}) + (Lw_{in} - Lw_{out}) \tag{7}$$

As a temperature sensor (Pt-100) is incorporated in the CNR4, air temperature was also recorded.

### 2.3.2 Scintillometer

A scintillometer is an instrument that consists of a transmitter that emits an electromagnetic

wave signal (with a specific wavelength $\lambda_s$) to a receiver, which records the intensity variations of this signal (Yee et al., 2015). The variations in signal intensity are caused by the fluctuations of the refractive index of air ($n$) along a propagation path (see Figure 5), because of eddies created by variations of temperature, humidity, or pressure in the Planetary Boundary Layer.



The magnitude of variations of a scalar, the refraction index of air $n$ for this case, can be
described by the structure function parameter of the scalar $C_n{}^2$.

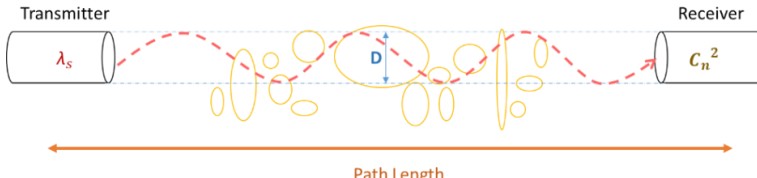

**Figure 5. Operational principle of a scintillometer.**

In this experiment, a Large Aperture Scintillometer (LAS) MKI produced by Kipp&Zonen®
(Kipp & Zonen, 2007) was installed at the highest levels of the BGW. The path length range of
a LAS is 250 m up to 4.5 km. However, shorter distances between 100 m to 1 km can be
measured if the aperture diameter (D) is reduced to 10 cm by a diaphragm. Another
specification of the LAS is the $\lambda_s$, which corresponds to 850 nm. LAS MKI provides spatially
average measurements of $C_n{}^2$. Indeed, LAS MKI measures the variance of the intensity
fluctuations natural logarithm $(\sigma_{lnI}^2)$, which is related to the path-average refractive index
structure parameter $C_n{}^2$, as follows:

$$C_n{}^2 = 1.12\sigma_{lnI}^2 D^{7/3}R^{-3} \tag{8}$$

where, $D$ is the aperture diameter of both transmitter and receiver, and $R$ is the path-length
between both LAS units.


When the intensity of $C_n{}^2$ is too elevated, the signal becomes saturated and the intensity
fluctuations of the scintillometer $\sigma_{ln(I)}^2$ are no longer proportional to $C_n{}^2$. This phenomenon is
known as the saturation effect. To avoid this, the pathlength, the aperture diameter and the
wavelength are used to evaluate the criterion of $C_n{}^2$ for saturation-free conditions, with the
formula $C_n{}^2 < 0.18D^{5/3}R^{-8/3}\lambda_s^{2/6}$.

The spatial fluctuations of $n$ along the path-length are associated with fluctuations of
temperature $(T)$, humidity $(Q)$, and to a lesser extent, air pressure $(P)$. Hence, the structure
parameter $C_n{}^2$ can be related to the thermodynamic structure parameters of air temperature $C_T{}^2$,
humidity $C_Q{}^2$ and the covariant term $C_{TQ}$ (and ignoring pressure fluctuations):

$$C_n{}^2 = \frac{A_T{}^2}{\overline{T}^2}C_T{}^2 + \frac{2A_T A_Q}{\overline{TQ}}C_{TQ} + \frac{A_Q{}^2}{\overline{Q}^2}C_Q{}^2 \tag{9}$$

where, $A_T$ and $A_Q$ are constants, functions of the beam wavelength and $\overline{T}$ is the mean value of
air temperature, and $\overline{Q}$ is the mean humidity.

Nevertheless, as demonstrate by Moene (2003) $C_T{}^2$ can be considered as directly proportional
to $C_n{}^2$ as this is more influenced by the temperature that the humidity:

$$C_n{}^2 \approx \frac{A_T{}^2}{\overline{T}^2}C_T{}^2 \tag{10}$$



From $C_T{}^2$, it is then possible to estimate $Qh$ applying the Monin-Obukhov Similarity Theory (MOST), jointly with additional meteorological data: air temperature, relative humidity, wind speed and air pressure (Kipp & Zonen, 2007). MOST derived the universal dimensionless relationship for $C_T{}^2$:

$$\frac{C_T{}^2 (z_{LAS}-d)^{\frac{2}{3}}}{T_*{}^2} = f_T \left( \frac{z_{LAS}-d}{L_{MO}} \right) \tag{11}$$

and

$$T_*{}^2 = \frac{C_T{}^2 (z_{LAS}-d)^{2/3}}{f_T \left( \dfrac{z_{LAS}-d}{L_{MO}} \right)} \tag{12}$$


where $d$ is the zero-displacement height (the height at which the mean velocity is zero due to large obstacles such as buildings/canopy), $z_{LAS}$ is the effective height of the scintillometer beam above the surface, $L_{MO}$ is the Monin-Obukhov length, $T_*$ is the temperature scaling variable and $f_T$ corresponds to the universal stability function.


In this way, from $C_n{}^2$ measurements, meteorological and terrain parameters, and the application of MOST relationships, $T_*$, $L_{MO}$ and the friction velocity $u_*$ are iteratively estimated. Finally, $Qh$ can be computed as follows:

$$Qh = -\rho_a C_p u_* T_* \tag{13}$$

where, $\rho_a$ is air density (~1.12 kg/m³ at sea level) and $C_p$ is the specific heat capacity of air at constant pressure (~1005 J/K kg).

### 2.3.3 Thermocouples

According to the Fourier's law, the conduction of heat flux into the soil is linearly proportional to the soil temperature gradient $\partial T/\partial z$ of the soil layer (expressed in K/m), and the capacity of
the soil to transfer heat, property known as soil thermal conductivity $k$ (W/m K):

$$Qg(z,t) = -k\frac{\partial T}{\partial z} \tag{14}$$

During the energy balance measurement campaigns, four thermocouples were placed in the BGW substrate, between the LAS receiver and the CNR4. They were vertically separated from
1 to 2 cm, aiming to estimate the soil thermal gradient as a direct application of the one-dimensional form of Fourier's Law, and quantify the heat flux transfer ($Qg$) by means of Eq. (14).

The experimental data of temperature gathered by the thermocouples placed in the deepest and
superficial location of the soil ($z_1 = 2$ and $z_4 = 6$ cm, respectively) as shown in Figure 6, was used to calculate the temperature gradient.



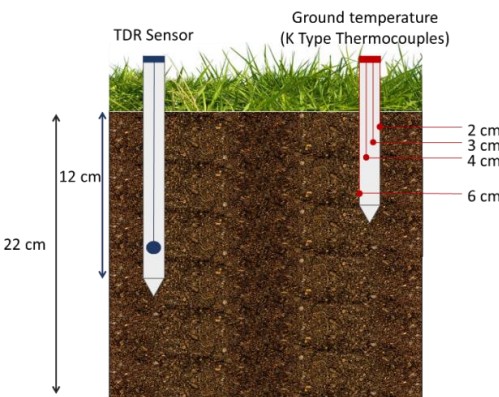

**Figure 6. TDR sensor and K type thermocouple set-up on the BGW.**

Since $k$ is a function of several factors, such as soil density, porosity, water content or thermal
conductivity of individual particles (Vera et al., 2018), there is a great difficulty in obtaining
accurate $k$ measurements under field conditions. In this experiment, a value of $k$ was set
according to a literature review (Vera et al., 2017), which is based on soil moisture conditions
of green roofs.

As already mentioned, two additional thermocouples were used with the evapotranspiration
chamber (see below) to measure the air temperature in and out of the chamber.

### 2.3.4 TDR water content sensor

The space-time variability of local water content and soil temperature ($T_{soil}$) was monitored on
the BGW by means of a wireless network of 16 CWS665 sensors (produced by Campbell
Scientific®) placed in different locations of the roof. The sensors use Time Domain
Reflectometry (TDR) technique to measure the propagation time of an electromagnetic (EM)
pulse. This pulse is applied to a pair of 12-cm metallic rods inserted into the soil. The time
necessary of the incident EM to reach the end of the rods and its reflection will depend on the
dielectric permittivity ($k_a$) of the soil. An empirical universal relationship between $k_a$ and the
volumetric water content ($VWC$) for a homogeneous mineral soil was established by Topp et
al. (1980):

$$VWC = -5.3 \times 10^{-2} + 2.92 \times 10^{-2}k_a - 5.5 \times 10^{-4}k_a^2 + 4.3 \times 10^{-6}k_a^3 \quad (15)$$

From $VWC$ (m³/ m³) measurements, the variations of soil water content were determined as
follows:

$$\Delta S = \frac{\Delta VWC}{\Delta t}z_s \quad (16)$$

where, $z_s$ the soil layer thickness and $\Delta t$ is the time step of soil water storage variation.

Then, ET in the BGW was deduced from the Eq. (5) and Eq. (16). It represents the height of
water loss by soil evaporation and plant transpiration over a dry period.

$$ET = -\Delta S = -\frac{\Delta VWC}{\Delta t}z_s \quad (17)$$




### 2.3.5 Gaz analyser

The concentration of absolute humidity within the evapotranspiration chamber was monitored through a LI-COR 7500 $CO_2/H_2O$ Gas Analyser (by LI-COR®, see Li-COR Biosciences, 2017 for details).


The measurement consists of a broadband infrared light source, band-pass filters to select a wavelength range that spans absorption lines for $CO_2$ and water vapor, and a detector. Light is absorbed by $CO_2$ and $H_2O$ in the light path, and the reduced intensity observed by the detector is a nonlinear function of the molar concentration of $CO_2$ and $H_2O$. The LI-7500 is an open-

path analyser that has a sample cell in open air.

Concentration data of $H_2O$ in mmol/mol measured by LI-COR 7500 was directly converted in g/m$^3$ for the deduction of ET from Eq. (3).

### 2.4 Monitoring campaigns

Due to technical and safety conditions, the measuring equipment was not permanently implemented on the BGW. In consequence, some daily monitoring campaigns were carried out during the 2018-2020 summers.

The LAS MKI was implemented on the BGW over some average periods of 7 hours. Each

scintillometer unit (transmitter and receiver) was situated on the highest points of the roof separated about approximately 100 m (see Figure 7). Diaphragms for short range applications were placed in front of the units, reducing the aperture diameter $D$ from 15 to 10 cm. Since the path length and the height of the LAS units are the only variables than can be modified to maintain $C_n{}^2$ below the saturation criterion, the transmitter and receiver heigh ($z_T$ and $z_R$) were

adjusted to respect $C_n{}^2 < 1.7 x 10^{-10}\ m^{-2/3}$ over the BGW.

Additional parameters related to the terrain, such as $z_0$ (roughness length) and $d$ (zero-displacement height), were adopted following the LAS MKI manual (Kipp & Zonen, 2007). For $z_0$, the BGW was considered as a rough terrain with open landscape and obstacles separated

by $\sim15h$ ($h$ being the crop height), while $d$ was considered negligible since the roughness elements of the BGW are not closely packed.

The radiometer was placed close to the LAS receiver at a height of 1.5 m (Figure 7). Additional necessary meteorological data, such as wind speed and direction, relative humidity, and air

pressure were not available in-situ. In consequence, they were gathered from the Orly Airport weather station, from the national meteorological service of France (Météo-France, https://meteofrance.com/), located 50 km from the BGW. The timestep of the dataset was 3 hours.





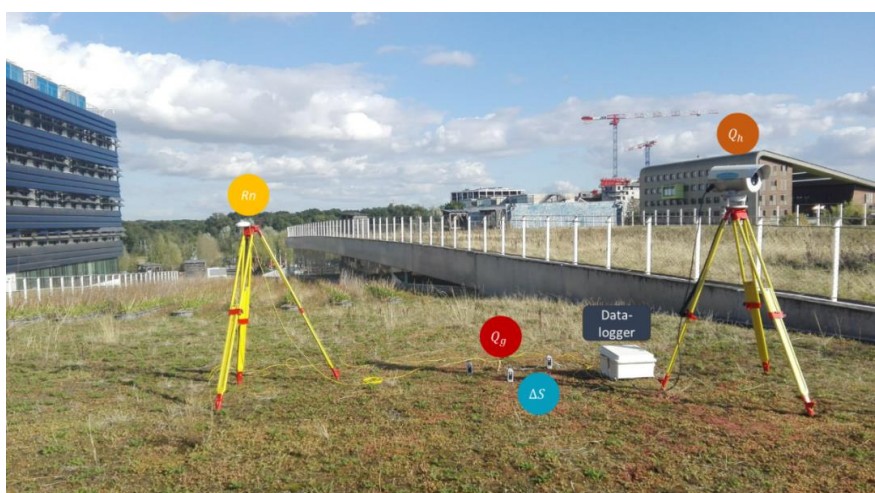

**Figure 7. Implementation of CNR4 (yellow), LAS MKI-receiver (orange), Type K Thermocouples (red) and TDR sensors (cyan) on the BGW.**

During the monitoring campaign of summers 2018 and 2019, the evapotranspiration chamber was placed over the ground of the BGW for two minutes every ten minutes to monitor the variation of absolute humidity, carbon dioxide, net radiation, and temperature. It was accompanied by 2 thermocouples (one inside the chamber, one outside) and by a radiometer placed inside the chamber. These sensors were placed about 20 cm above the ground surface.

### 2.4 Data processing

The energy balance measurements from the LAS MKI, CNR4 and thermocouples were collected with a CR3000 Datalogger (from Campbell Scientific®). A program was created through the software LoggerNet (i.e., from the same Datalogger manufacturer) to communicate with the sensors, and to collect and store the data. The LAS MKI manufacturer provides the software EVATION (Kipp & Zonen B.V, 2015) to users. It performs estimations of $Qh$ by using the monitored data and those collected from Météo-France, and applying the iterative procedure presented previously. The resulting data are synthetized in a unique file (YYYY_MM_DD_Data_SEB.csv). Each line corresponds to a time step of 10 minutes for which every information is recorded (the values are separated by a comma). The different columns are detailed in Annex.

The sensors associated with the evapotranspiration chamber (gas analyser, radiometer, thermocouples) were connected to a CR800 Datalogger (from Campbell Scientific®) to collect measurements every second. A python script was written to verify the adequate correspondence between the steps of installation and removal of the chamber with the increase in the absolute humidity rate, and determines an evapotranspiration value for each handling. The resulting data are synthetized in a unique file (YYYY_MM_DD_EvapotranspirationData.csv) and the contain of each column is detailed in Annex.

As already presented in details in Versini et al. (2020), $k_a$ and $T_{soil}$ data from TDR sensors are transmitted to a CWB100 wireless base, which transfers them to a CR6 Datalogger (from Campbell Scientific®). The data are collected and stored every night on the HM&Co server.



Here, this data has been gathered with the help of a Python script to create 10-minute resolution time series for every campaign. They are stored in a specific file (YYYY_MM_DD_Data_WB.csv), in which each line corresponds to a time step for which both water content and temperature data are recorded (see Annex).

## 3-      Data availability

The daily monitoring campaigns were carried out during summers 2018, 2019, and 2020. Unfortunately, not all measuring equipment was used at the same time during this period. The data measured by the Scintillometer in 2018 was not considered, as the sensor had to be recalibrated. Moreover, the transpiration chamber was out of order in 2020. Finally, the three measurement methods (energy balance, water balance and transpiration chamber) were
simultaneously operational only in 2019. Nevertheless, all daily campaigns measurements are made available and presented below. This data set (Versini et al., 2023a) is available for download from the following web page: https://doi.org/10.5281/zenodo.8064053.

### 3.1 Presentation of the daily campaigns

Table  presents the different daily campaigns carried out during the 2018-2020 time period.  For each date, the availability of the data regarding the 3 measurement methods is mentioned. In addition, some indications concerning the hydro-meteorological conditions are reported. Antecedent humidity was estimated for each date from the precipitation measured during the previous week by the Taranis platform (see Gires et al., 2018). Weather condition was visually
reported and classified in 3 categories (sunny, cloudy and variable when clouds occasionally darkened the sky). The ranges of measured air temperature and net radiation are also mentioned.

**Table 1. Presentation on the daily campaigns and availability of the data**

|  | ETP chamber | SEB | WB | Antecedent humidity | Weather conditions | T [min-max] | Rn [min-max] |
|---|---|---|---|---|---|---|---|
| 20/06/2018 | X |  | X | < 1 mm | Sunny | 27-32 °C |  |
| 21/08/2018 | X |  | X | < 1 mm | Cloudy | 24-32 °C |  |
| 26/09/2018 | X |  | X | 9 mm | Sunny | 22-25 °C |  |
| 27/09/2018 | X |  | X | 9 mm | Sunny | 20-30 °C |  |
| 10/07/2019 | X | X | X | < 1 mm | Sunny | 25-38 °C | 380-700 W/m² |
| 08/08/2019 | X | X | X | 17 mm | Variable | 19-34 °C | 250-750 W/m² |
| 23/08/2019 | X | X | X | 18 mm | Sunny | 18-30 °C | 220-650 W/m² |
| 29/08/2019 | X | X |  | < 1 mm | Variable | 20-28 °C | 120-650 W/m² |
| 30/09/2019 |  | X | X | 20 mm | Variable | 12-20 °C | 150-600 W/m² |
| 02/10/2019 |  | X | X | 17 mm | Variable | 10-19 °C | 50-600 W/m² |
| 03/10/2019 |  | X | X | 14 mm | Cloudy | 7-17 °C | 50-550 W/m² |
| 01/07/2020 |  | X | X | 7 mm | Cloudy | 21-25 °C | 50-600 W/m² |
| 08/07/2020 |  | X | X | 1 mm | Variable | 22-34 °C | 200-800 W/m² |
| 09/07/2020 |  | X | X | < 1 mm | Sunny | 24-36 °C | 150-750 W/m² |
| 17/07/2020 |  | X | X | 4 mm | Cloudy | 20-32 °C | 200-750 W/m² |
| 22/07/2020 |  | X | X | 4 mm | Sunny | 22-35 °C | 300-700 W/m² |
| 23/07/2020 |  | X | X | < 1 mm | Sunny | 23-36 °C | 100-700 W/m² |



### 3.2 Presentation of the available data set and scripts

For each campaign day, a specific folder has been constituted (YYYY_MM_DD, 2019_07_10 for example). Every folder contains 3 datafiles (when they exist) regrouping the measured data for the 3 different methods : Surface Energy Balance, Transpiration chamber rand Water Balance respectively (YYYY_MM_DD_Data_SEB.csv, YYYY_MM_DD_ETCH.csv, YYYY_MM_DD_Data_WB.csv).

For each method, a Python script has also been provided to select the monitoring campaign and the associated data, transform this data in physical variables, and carry out some first analysis. An additional script is proposed to study and represent the soil, surface and air temperatures computed from the different sensors. The 4 Python scripts are presented in details in the next sub-sections.

### 3.1.1 Read_Data_SEB.py

This Python script aims to read the data measured during the SEB campaigns. It is structured as follow:

- Campaign selection: This part can be modified to select the monitoring campaign to be considered (13 dates are proposed).

- Data selection and transformation: the associated data file corresponding to the measurement method is opened and read. The conduction heat flux is computed by using soil temperature data and Eq. (14). The latent heat flux is also deduced from the energy balance equation following Eq. (2).

- Representation of the computed data: A figure is proposed to illustrate and analyze the produced data. It represents the different components of the Surface Energy Balance: net radiation computed with the radiometer ($Rn$), conduction heat flux computed with the thermocouples ($Qg$), sensitive heat flux computed with the scintillometer ($Qh$), and the latent heat flux deduced from the previous component ($Qe$).

### 3.1.2 Read_Data_ETPCH.py

This Python script aims to read the data measured during the evapotranspiration chamber campaigns. It is structured as follow:

- Campaign selection: This part can be modified to select the monitoring campaign to be considered (8 dates are proposed).

- Data selection and transformation: the associated data file corresponding to the measurement method is opened and the evapotranspiration deduced from Eq. (3) is read.

- Representation of the computed data: the punctual measurements carried out by the evapotranspiration chamber are depicted in a specific figure.



### 3.1.3 Read_Data_WB.py

This Python script aims to read and transform the data measured by the TDR water content sensors. It is structured as follow:

- Campaign selection: This part can be modified to select the monitoring campaign to be considered (16 dates are proposed).

- Data selection and transformation: Regarding the water balance method, the dielectric constant measured by the 16 TDR sensors are converted in volumetric water content by using the Topp equation (Eq. 15). It is possible to smooth this data by using a moving window. Then, the difference between 2 consecutive water content measures is computed to estimate water loss. Finally, these values are transformed in latent heat flux by using Eq. (17) and an average value for all available TDR sensors is computed.

- Representation of the computed data: A first figure represents the volumetric water content computed from the Topp equation for the 16 TDR sensors. A second figure depicts the average resulting latent heat flux deduced from the TDR sensors.

### 3.1.4 Read_Temperature.py

This Python script aims to read and compare the different temperature data (in the soil, on the surface, in the air) measured during the different protocols. It is structured as follow:

- Campaign selection: This part can be modified to select the monitoring campaign to be considered (17 dates are proposed).

- Data selection: Depending on data availability, the data files corresponding to the 3 different methods are read. The temperature measurements are specifically extracted. They correspond to: soil temperatures measured from the thermocouples placed in the substrate layer, air temperature (from the CNR4 radiometer, and thermocouples in and outside the evapotranspiration chamber), and surface temperature from the TDR sensors.

- Representation of the computed data: Theses temperature data measured by the different sensors are depicted. The legend, the scale, and the time window are adapted depending on the availability of the data.

### 3.3 Presentation of an example

To illustrate the provided data and scripts, the daily campaign of the 8th August 2019 has been chosen. It has the advantage to be one of the days for which the 3 campaigns were simultaneously carried out. The weather condition for this campaign was considered as "Variable" with a blue sky crossed by some clouds during the day. It is clearly visible on Figure 8 where the different components of the SEB are represented. The net radiation $Rn$ fluctuates during the afternoon reaching more than 700 W/m$^2$ around 13:30 and decreasing to 200 W/m$^2$ at 16:00. The sensible flux $Qh$ fluctuates also, but in a smaller range, between 60 to 320 W/m$^2$. The conduction flux $Qg$ is the weakest flux, reaching 100 W/m$^2$ at the end of the measurement campaign. It illustrates the inertia properties of the substrate, whose vertical gradient of temperature increases during the day (see also Figure 11). The resulting latent heat flux $Qe$ roughly follows $Qh$ and varies between 100 to 600 W/m$^2$.

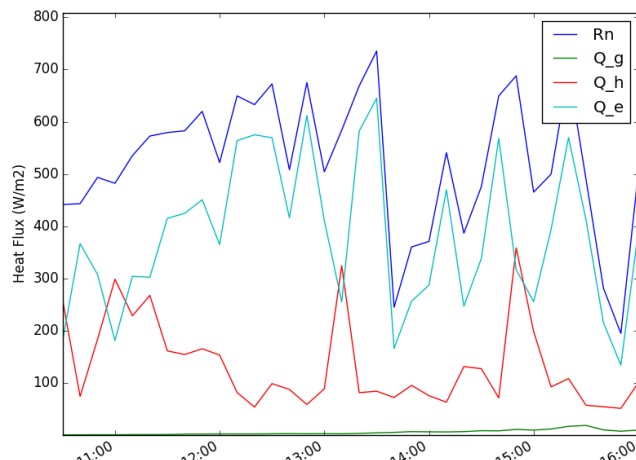

**Figure 8. Surface Energy Balance components estimated for the 8th August 2019 campaign**

The evapotranspiration estimated with the chamber fluctuates also during the whole campaign (see Figure 9), but in a quite different range of values (from 60 to 140 W/m$^2$). The low values occurred around 14:00 like for the SEB campaign. This can be explained by the presence of clouds in the sky, resulting in less radiative flux received and therefore less evapotranspiration. Although the order of magnitude is similar, the difference between the evapotranspiration values estimated using the two methods may be questionable. One reason for this difference can obviously be a measurement inconsistency, but it should also be remembered that these methods are carried out at two different spatial scales.

**Figure 9. Latent Heat flux estimated by the transpiration chamber for the 8th August 2019**

Finally, the evapotranspiration estimated from the water balance assessed by the TDR sensors network is represented in Figure 10. The order of magnitude of the evapotranspiration estimates, as well as their range, is similar to the two previous ones. In this case, the fluctuation can be explained by the low difference between two consecutive values, that can be sometimes



negative. This is due to the resolution of the sensor which is not actually adapted to the selected time step (10 minutes).

Moreover, the 16 TDR sensors do not work together due to punctual dysfunction. Although they represent the same dynamic, the sensors show a significant variability in terms of absolute value. These differences illustrate the heterogeneity of this granular substrate in terms of pore size distribution and hydrological behavior. It is worth noting that the water content value computed from the Topp equation (see Eq 15) could also be negative. Indeed, this formula was

developed for traditional soils, significantly different from granular substrates as that implemented on the Blue Green Wave. This difference is more pronounced during dry periods, where very low water content values are expected.

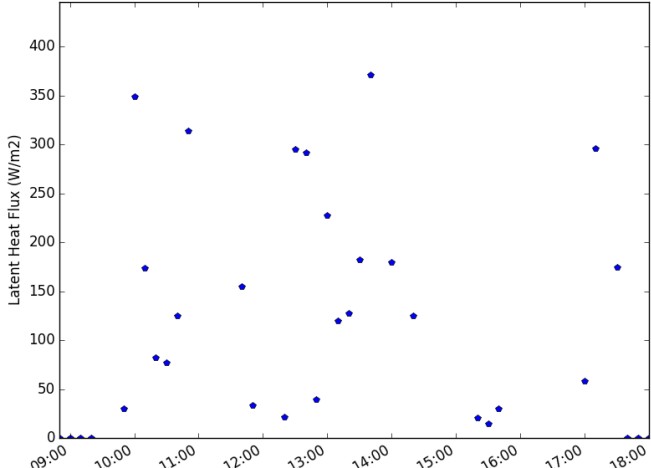

**Figure 10. Evapotranspiration estimated from the Water Balance for the 8th August 2019**

The different temperatures measured during the daily campaign are represented in Figure 11. The different locations where these measurements were made are clearly distinguishable. The soil temperatures (T_1, T_2, T_3, T_4) follow the same trend and increase during the day. The vertical profile is also coherent with the lowest values at depth and the highest close to the

surface. The difference, which was only a few tenths of a degree at the beginning of the day, can reach 1°C at the end of the campaign. It illustrates the thermal inertia of the substrate. The surface temperature measured by the TRD sensors is less variable, ranging between 26 and 29°C. In the shade of the sensor, the measurement does not take into account direct radiation from both the sun and the atmosphere. Finally, the air temperature measurements are quite

similar from one sensor to another. The most reliable ones seem to be the measurements performed by the thermocouples inside and outside the chamber. They fluctuate together and match very well. Implemented higher (1.5 m instead of 20 cm for the thermocouples), the temperature measurement carried out by the CNR4 is also in accordance, and seems not to be influenced by the internal heat of the sensor.


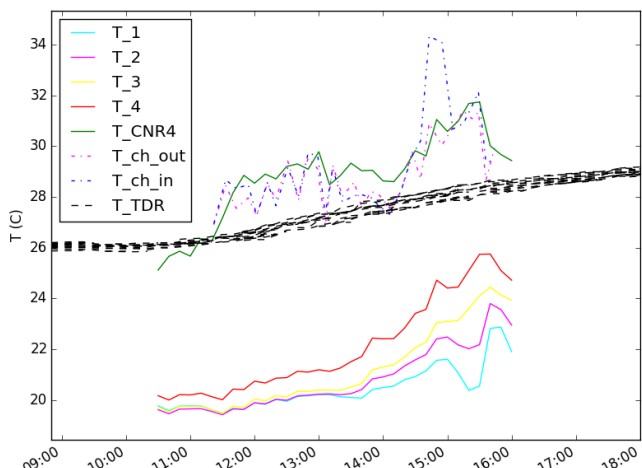

**Figure 11. Comparison of the different temperature measures for the 8th August 2019**

## 4- Conclusion

This paper presents the thermo-hydric data collected by 3 different methods to assess the
evapotranspiration process representing 3 different spatial scales. The first one is based on the
Surface Energy Balance (large scale), the second one is carried out by an evapotranspiration
chamber and absolute humidity measurement (small scale), and the third one is based on the
Water Balance evaluated during dry periods (punctual scale). In addition to these
evapotranspiration estimates, several hydro-meteorological data are made available: incident
and reflected solar and atmospheric radiations, substrate, surface and air temperature, water
content, structure parameter of the refractive index of air… This data was collected during daily
campaigns for 3 consecutive summers (2018, 2019, 2020). The presented dataset is available
for download free of charge from the following webpage:
https://doi.org/10.5281/zenodo.8064053

It is provided by the Hydrology, Meteorology and Complexity laboratory from Ecole des Ponts
(HM&Co-ENPC). The following references should be cited for every use of the data: Versini
et al., 2023a, Versini et al., 2023b).

These measurements are useful to study the ability of nature-based solutions (NBS) like green
roofs to locally cool and mitigate urban heat islands. Evapotranspiration represents the key-
process conducting this cooling effect. Therefore, the presented data can be used to increase
knowledge concerning the estimation of evapotranspiration through the quantification of hydro-
meteorological variables. This data could also be used to develop, calibrate and/or validate
microclimate models (as Solene-Microclimat (Malys et al., 2014) or ENVI-met (Liu et al.,
2021)) taking into account NBS in their applications.

The HM&Co-ENPC lab still pursues his research activities on the multi-scale ecosystem
services assessment of NBS, and the use of the Blue Green Wave as a demonstrator
experimental site. The monitoring campaigns will continue with a new scintillometer (Scintec
SLS20) to study the spatio-temporal variability of the involved processes, and particularly the
sensible heat flux of great importance in the energy balance. Combined with the additional





measurement methods, the objective is to improve the assessment of the evapotranspiration flux through scales.


This paper focussed on urban heat islands mitigation appears as a complement of a previous data paper focussed on hydrological data and the use of NBS to manage stormwater (Versini et al, 2019). This data contributes to the improvement of the Multi-Hydro platform (El-Tabach et al., 2009, Qiu et al., 2021). Initially developed for rainfall-runoff modelling in urban

environments, a first coupling with Solene-Microclimate was carried out during the EVNATURB project (https://hmco.enpc.fr/portfolio-archive/evnaturb/). It aimed to take benefit of the distributed modelling approach to propose a coupled modelling of thermic and hydrological processes. Some efforts remain to improve this coupling and to reproduce evapotranspiration flux through spatial scales (from the infrastructure to the district).


### Author contribution

Pierre-Antoine Versini supervised the study, reviewed, and wrote a large part of the manuscript and the Python scripts. Leydy Alejandra Castellanos-Diaz wrote also a significant part of the manuscript and the scripts, carried out the measurement's campaigns and the collection of the

data, and also participated to the review of the paper. David Ramier and Ioulia Tchiguirinskaia collaborated to the study supervision and the review process.

### Competing interests

The authors declare that they have no conflict of interest.


### Acknowledgments

This work has been supported by the ANR EVNATURB project (ANR-17-CE22- 0002-01) dealing with the evaluation of ecosystem performance for renaturing urban environments. It has also been supported by the Academic Chair "Hydrology for Resilient Cities", a partnership be-

tween École des Ponts ParisTech and the Veolia company. The authors particularly thank Georges Najjar from the University of Strasbourg for having providing the MKI scintillometer, the Cerema for the evapotranspiration chamber and the Kipp & Zonen company for their support concerning the sensors implementation.

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

            Evapotranspiration Fluxes on Green Roofs: A Comparative Analysis of Observational
            Methods. STOTEN, In press., 2023

       Poë, S., Stovin, V., Berretta, C.: Parameters influencing the regeneration of a green roof's
retention capacity via evapotranspiration, Journal of Hydrology 523, 356–367.
            https://doi.org/10.1016/j.jhydrol.2015.02.002, 2015

       Qiu, Y., da Silva Rocha Paz, I., Chen, F., Versini, P.-A., Schertzer D, Tchiguirinskaia I:
            Space variability of hydrological responses of Nature-Based Solutions and the
            resulting uncertainty, Hydrology and Earth System Sciences, 25(6), 3137-3162.
https://doi.org/10.5194/hess-25-3137-2021, 2021

       Ramier, D., Chollet, J., Berthier, E., Sabre, M., Tétard, Y., Flori, J.-P., Bouyer, J.: Mesure de
            l'évapotranspiration à petite échelle spatiale : applications aux toitures végétalisées,
            Journée Scientifique du SIRTA, Palaiseau, France.
            https://sirta.ipsl.polytechnique.fr/documents/JSS2015/posters/JSS2015_poster-
A5_Ramier.pdf, 2015

       Rana, G., Katerji, N.: Measurement and estimation of actual evapotranspiration in the field
            under Mediterranean climate: a review, European Journal of Agronomy 13, 125–153.
            https://doi.org/10.1016/S1161-0301(00)00070-8, 2000

       Sharma, A., Conry, P., Fernando, H.J.S., Hamlet, A.F., Hellmann, J.J., Chen, F. Green and
cool roofs to mitigate urban heat island effects in the Chicago metropolitan area:
            evaluation with a regional climate model, Environmental Research Letters 11, 064004.
            https://doi.org/10.1088/1748-9326/11/6/064004, 2016a

       Sharma, A., Conry, P., Fernando, H.J.S., Hamlet, A.F., Hellmann, J.J., Chen, F.: Green and
            cool roofs to mitigate urban heat island effects in the Chicago metropolitan area:
evaluation with a regional climate model, Environ. Res. Lett. 11, 064004.
            https://doi.org/10.1088/1748-9326/11/6/064004, 2016b

       Stanic, F., Cui, Y.J., Versini, P.-A., Schertzer, D., Tchiguirinskaia, I.: A device for the
            simultaneous determination of the water retention properties and the hydraulic



conductivity of an unsaturated green-roof material, Geotechnical Testing Journal,
2019

Stovin, V., Vesuviano, G., Kasmin, H.: The hydrological performance of a green roof test bed
under UK climatic conditions, Journal of Hydrology 414–415, 148–161.
https://doi.org/10.1016/j.jhydrol.2011.10.022, 2012

Tabares, P.C., Srebric, J.: A heat transfer model for assessment of plant based roofing systems
in summer conditions, Building and Environment 49, 310–323.
https://doi.org/10.1016/j.buildenv.2011.07.019, 2012

Thorp, R.: Observations of heat, water vapor and carbon dioxide exchanges over a living roof
using eddy covariance, San Francisco State University, 2014

Topp, G.C., Davis, J.L., Annan, A.P.: Electromagnetic determination of soil water content:
Measurements in coaxial transmission lines, Water Resources Research 16, 574–582.
https://doi.org/10.1029/WR016i003p00574, 1980

Valayamkunnath, P., Sridhar, V., Zhao, W., Allen, R.G.: Intercomparison of surface energy
fluxes, soil moisture, and evapotranspiration from eddy covariance, large-aperture
scintillometer, and modeling across three ecosystems in a semiarid climate,
Agricultural and Forest Meteorology 248, 22–47,
https://doi.org/10.1016/j.agrformet.2017.08.025, 2018

Vera, S., Pinto, C., Tabares-Velasco, P.C., Bustamante, W.: A critical review of heat and
mass transfer in vegetative roof models used in building energy and urban enviroment
simulation tools, Applied Energy 232, 752–764.
https://doi.org/10.1016/j.apenergy.2018.09.079, 2018

Vera, S., Pinto, C., Tabares-Velasco, P.C., Bustamante, W., Victorero, F., Gironás, J., Bonilla,
C.A.: Influence of vegetation, substrate, and thermal insulation of an extensive
vegetated roof on the thermal performance of retail stores in semiarid and marine
climates, Energy and Buildings 146, 312–321.
https://doi.org/10.1016/j.enbuild.2017.04.037, 2017

Versini, P.-A., Stanic, F., Gires, A., Schertzer, D., Tchiguirinskaia, I.: Measurements of the
water balance components of a large green roof in the greater Paris area, Earth System
Science Data 12, 1025–1035. https://doi.org/10.5194/essd-12-1025-2020, 2020

Versini, P.-A., Castellanos-Diaz, L A, Ramier, D., and Tchiguirinskaia, I.: Evapotranspiration
evaluation by 3 different protocols on a large green roof in the greater Paris area -
DataSet https://doi.org/10.5281/zenodo.8064053, 2023a

Versini, P.-A., Castellanos-Diaz, L A, Ramier, D., and Tchiguirinskaia, I.: Evapotranspiration
evaluation by 3 different protocols on a large green roof in the greater Paris area, Earth
System Science Data, under review, 2023b

Voyde, E., Fassman, E., Simcock, R., Wells, J.: Quantifying Evapotranspiration Rates for
New Zealand Green Roofs, Journal of Hydrologic Engineering 15, 395–403.
https://doi.org/10.1061/(ASCE)HE.1943-5584.0000141, 2010

Wadzuk, B.M., Schneider, D., Feller, M., Traver, R.G.: Evapotranspiration from a Green-
Roof Storm-Water Control Measure, Journal of Irrigation and Drainage Engineering
139, 995–1003. https://doi.org/10.1061/(ASCE)IR.1943-4774.0000643, 2013a

Wadzuk, B.M., Schneider, D., Feller, M., Traver, R.G.: Evapotranspiration from a Green-
Roof Storm-Water Control Measure, J. Irrig. Drain Eng. 139, 995–1003.
https://doi.org/10.1061/(ASCE)IR.1943-4774.0000643, 2013b

Yang, J., Mohan Kumar, D. llamathy, Pyrgou, A., Chong, A., Santamouris, M., Kolokotsa,
D., Lee, S.E.: Green and cool roofs' urban heat island mitigation potential in tropical
climate, Solar Energy 173, 597–609. https://doi.org/10.1016/j.solener.2018.08.006,
2018





Yee, M.S., Pauwels, V.R.N., Daly, E., Beringer, J., Rüdiger, C., McCabe, M.F., Walker, J.P.
A comparison of optical and microwave scintillometers with eddy covariance derived
surface heat fluxes, Agricultural and Forest Meteorology 213, 226–239.
https://doi.org/10.1016/j.agrformet.2015.07.004, 2015

**Annexes**

Content of each line of the SEB datafile:
- time step expressed as YYYY- MM-DD HH:MM
- item number
- Internal temperature (°C)
- upper pyranometer (solar radiation)
- lower pyranometer
- External temperature (°C)
- External temperature (K)
- upper pyrgeometer (W/m$^2$)
- lower pyrgeometer (W/m$^2$)
- Net Solar radiation = (Upper pyranometer) - (Lower pyranometer) (W/m$^2$)
- Net Far Infrared radiation = (Upper pyrgeometer) - (Lower pyrgeometer) (W/m$^2$)
- Albedo = (Lower pyranometer) / (Upper pyranometer) (W/m$^2$)
- Net radiation = (Upper pyranometer + Upper pyrgeometer) - (Lower pyranometer + Lower pyrgeometer) (W/m$^2$)
- Soil temperature 1 (°C)
- Soil temperature 2 (°C)
- Soil temperature 3 (°C)
- Soil temperature 4 (°C)
- Relative humidity (%)
- Pressure (hPa)
- Wind velocity (m/s)
- Wind direction (°)
- Net radiation (W/m$^2$)
- Log of $Cn^2$ structure function PU$Cn^2$ (V)
- Sig_ PU$Cn^2$ (V)
- Received fluctuation intensity (Udemod) (mV)
- Variance of the Received fluctuation intensity Sig_Udemod (mV)
- Conduction heat flux (W/m$^2$)
- Sensitive heat flux (W/m$^2$)
- Bowen coefficient
- Estimated latent heat flux (W/m$^2$)

Content of each line of the evapotranspiration chamber datafile:
- time step expressed as YYYY- MM-DD HH:MM:SS
- estimation of the latent heat flux (W/m$^2$)
- Net radiation (W/m$^2$)



- External temperature (°C)
- Internal temperature (°C)

840    Content of each line of the TDR datafile:

- exact definition of the time step expressed in YYYY- MM-DD HH:MM
- item number
- volumetric water content (expressed as $k_a$) for the 16 TDR sensors
- temperature (expressed in °C) for the 16 TDR sensors

845