# Peer review of "Evapotranspiration evaluation by 3 different protocols on a large green roof in the greater Paris area"

_Earth System Science Data, 2023_

## Author Response (AR1)

**Response to reviewers**

The Authors would like to thank the Reviewers for having considered the manuscript and for their constructive comments. The Authors deeply appreciate the recommendations and acknowledge their importance for improving this data paper. Accordingly, the Authors have revised the manuscript carefully, clarified all highlighted issues, and answered all the related questions. All the performed modifications and corrections are explained below and were incorporated in the revised manuscript.

**Reviewer 1**

Specific Comments:

- I continue to have reservations about the sourcing of key meteorological data from some 50 km away from the test location (L372). The manuscript stresses the importance of understanding ET processes to mitigate UHI effects. However, there is no mention of the micro-meteorological phenomena that help to create an UHI effect which can occur at much smaller scales than 50 km. Please demonstrate the insensitivity of final ET values to these meteorological data or otherwise more adequately caution the reader of their influence on the results.

Indeed, micro-meteorological phenomena can have some influence on UHI creation. Wind only intervenes directly in the large scale evapotranspiration assessment by Surface Energy Balance, and more precisely in the sensible heat flux estimated by scintillometer. To understand the variables that most impact the iterative process of $Q$h calculation (Eq. 10, 11, 12), a sensitivity analysis through a Latin Hypercube Sampling (LHS) was conducted (see Castellanos, 2022).The Pearson's Correlation coefficient used as a sensitivity index in this analysis showed that the wind speed $U$ was moderately correlated to $Q$h (0.35). Moreover, for these particular cases, the monitoring campaigns occurred in relative no-wind conditions, for which we can assume very few differences between local data and Meteo France data. Note that, since these campaigns occurred, some anemometers have been implemented on the ENPC campus to have local wind measurments.

All this information has been added (P.11 L.384-391)

- I don't believe the presentation of the python scripts (section 3.2) to be helpful in demonstrating the importance/novelty of the dataset. I can appreciate these scripts are provided to aid accessibility, but this content is more appropriate in the README file that accompanies the data in the open access repository. Please consider removing and replacing with greater comparison of the obtained ET values from the three methods (where appropriate).

As mentioned in the guidelines for authors, "all material required to understand the essential aspects of the paper such as experimental methods, data, and interpretation should preferably be included in the main text". In consequence the description of the python scripts has be transferred in some Appendices as advised by HESSD. These descriptions contain some elements to explain the provided scripts to read and analyze the data, and how the equations presented in the manuscript are

used. In return, some elements have been added in the presentation of the example: P.15 L.582-585, P.15 L. 592-594.

**Reviewer 2**

While the Python scripts could be handy to some extent to get a quick impression of the provided data, they do not work in my environment (Python 3.11, Anaconda). In each script I get error messages. So, to the very least, it should be stated in which environment these scripts are expected to work free of errors. Please add the acronyms for each variable in the glossary in the annex.

Indeed, the Python scripts have been written and developed in Python 2.6. It has been indicated (P.14 L.458) as the acronyms (in an Appendix) in the revised version of the manuscript.

Detailed remarks:

p. 2 l. 87: EC is useful but only applicable to very large green roofs with sufficient fetch

We completely agree. As explained, EC can used for agricultural purposes on large crops. Some similar uses on green roofs require a very large asset. It has been indicated (P.2 L.88).

p. 9 l. 281: the term is usually called "zero-plane displacement height". And you might add that the velocity at this height + roughness length is zero according to the logarithmic wind profile.

"zero-displacement heigh" has been replaced by "zero-plane displacement height". Additional information about the consequences related to the assumption of the wind profile has also been added (P.9 L.282-284): "the zero-plane displacement height (the height to which the roughness length is added to define the height where the logarithmic wind profile is equal zero due to obstacles such as buildings/canopy)"

p. 9 l. 305: As it seems QE was estimated as a residual term. Therefore, QG should be calculated as close to the surface as possible, i.e. with z1 and z2.

We had chosen z1 and z4 to capture the temperature gradient in the substrate layer profile. It is possible to compute Qg closest to the surface by using z1 and z2. It has been indicated in the manuscript (P.9 L.307-309). The Python lets the possibility to choose the thermocouples in order to compute Qg.

p. 10 l. 311: which value was k set to? I suggest to implement a variable k in dependence on VWC according to Sailor and Hagos 2011 e.g. since k can vary by a factor of 2.

For now, we have proposed only two values for *k*, corresponding to dry condition (0.15 W/mK) and wet condition (0.85 W/mK) regarding Vera et al. (2017). It is now indicated in the manuscript (P.10 L.322-324) as it was only mentioned in the Python script. This range is similar to that illustrated in Sailor and Hagos (2011). Nevertheless, it can be modified in the Python script to better take into account this variability.

p. 11 l. 372: I presume it was interpolated to match the time step of the scintillometer measurements? Which method was used for that?

This wind measurement a clearly a weak point in our study (see answer to reviewer 1). In the absence of a local measure, we have used the wind data from the Orly Airport weather station, located 50 km from the BGW. As this dataset is characterized by a resolution of 3 hours, we used constant values in this interval to match with the scintillometer time step. All this information has been added (P.11 L.384-391)

Figure 10: I recommend to use lines (no smoothing) here as well for easier comparison.

The figure has been modified for a question of coherency.

p. 23 l. 800: "line" should be "column" I guess.

Corrected! Sorry for this mistake.